# Over-Winter Survival and Nest Site Selection of the West-European Hedgehog (*Erinaceus europaeus*) in Arable Dominated Landscapes

**DOI:** 10.3390/ani10091449

**Published:** 2020-08-19

**Authors:** Lucy E. Bearman-Brown, Philip J. Baker, Dawn Scott, Antonio Uzal, Luke Evans, Richard W. Yarnell

**Affiliations:** 1Department of Animal & Agriculture, Hartpury University, Gloucestershire GL19 3BE, UK; 2School of Biological Sciences, University of Reading, Reading RG6 6AH, UK; p.j.baker@reading.ac.uk (P.J.B.); L.C.Evans@pgr.reading.ac.uk (L.E.); 3School of Life Sciences, Keele University, Staffordshire ST5 5BG, UK; d.scott@keele.ac.uk; 4School of Animal, Rural & Environmental Sciences, Nottingham Trent University, Southwell, Nottinghamshire NG25 0QF, UK; antonio.uzal@ntu.ac.uk (A.U.); richard.yarnell@ntu.ac.uk (R.W.Y.)

**Keywords:** *Erinaceus europaeus*, farmland, habitat fragmentation, hedgerow, hibernacula, hibernation, mammal, nest

## Abstract

**Simple Summary:**

Hedgehogs (*Erinaceus europaeus*) have declined markedly in the UK in recent decades. One key stage that could affect their population dynamics is the annual winter hibernation period. Therefore, we studied two contrasting populations in England to examine patterns of winter nest use, body mass changes and survival during hibernation. On average, animals at both sites weighed the same prior to, and used the same number of nests, during hibernation. There was a marked difference in survival rates between the two sites, but no animals died during hibernation; all deaths occurred prior to or after the hibernation period, mainly from predation or vehicle collisions. Hedgehogs consistently nested in proximity to some habitats (hedgerows, roads, woodlands) but avoided others (pasture fields); the use of other habitats (arable fields, amenity grassland, buildings) varied between the two sites. These data suggest: (i) that hibernation was not a period of significant mortality at either site for individuals that had attained a sufficient weight (>600 g) in autumn; but that (ii) habitat composition did significantly affect the positioning of winter nests, such that different land management practices (historic and current) could influence hibernation success.

**Abstract:**

The West-European hedgehog (*Erinaceus europaeus*) has declined markedly in the UK. The winter hibernation period may make hedgehogs vulnerable to anthropogenic habitat and climate changes. Therefore, we studied two contrasting populations in England to examine patterns of winter nest use, body mass changes and survival during hibernation. No between-site differences were evident in body mass prior to hibernation nor the number of winter nests used, but significant differences in overwinter mass change and survival were observed. Mass change did not, however, affect survival rates; all deaths occurred prior to or after the hibernation period, mainly from predation or vehicle collisions. Hedgehogs consistently nested in proximity to hedgerows, roads and woodlands, but avoided pasture fields; differences between sites were evident for the selection for or avoidance of arable fields, amenity grassland and buildings. Collectively, these data indicate that hibernation was not a period of significant mortality for individuals that had attained sufficient weight (>600 g) pre-hibernation. Conversely, habitat composition did significantly affect the positioning of winter nests, such that different land management practices (historic and current) might potentially influence hibernation success. The limitations of this study and suggestions for future research are discussed.

## 1. Introduction

Agricultural intensification and climate alteration are two anthropogenic processes that have profound impacts on natural ecological systems [1,2,3,4,5,6,7]. The effects arise from a wide range of underlying causal factors including: habitat destruction, fragmentation and degradation [8,9]; the introduction of livestock, diseases and non-native biological control agents [10,11,12,13,14]; the management of wildlife where they conflict with human interests [15,16,17,18]; the application of chemical biocides [19]; and changes in the phenology of key biological events [20,21]. Collectively, these factors have led to the decline, extirpation and extinction of large numbers of species [22,23,24,25,26,27], but also increases in the abundance and geographic range of others (e.g., [28,29]).

One group of species that might be expected to be particularly affected by agricultural practices and changing climatic conditions are hibernators [30,31,32,33]. Hibernation has typically evolved to enable species to survive periods of prolonged food shortages by dramatically reducing levels of energy expenditure [34,35]. One consequence of this is that hibernating species tend to have slower reproductive rates [36], potentially increasing their long-term vulnerability to human activities.

The West-European hedgehog (*Erinaceus europaeus*, hereafter ‘hedgehog’) is a medium-sized (<1.2 kg) insectivorous mammal found from the Iberian Peninsula and Italy northwards into Scandinavia [37]. In Britain, hedgehogs were historically found throughout a broad range of agricultural landscapes [38,39,40,41], but rural populations have declined markedly in recent decades [42,43,44]. Consequently, hedgehogs are now increasingly found within areas of human habitation in this country [45,46,47] and elsewhere [48,49]. Associated with this decline has been a substantial reduction in the availability [50] and quality [51,52,53] of hedgerows, an important habitat for foraging [54], dispersal [55] and refuge [56], and a substantive increase in the numbers of badgers (*Meles meles*) [57,58], an intra-guild predator [59].

During hibernation, hedgehogs face specific challenges. First, they need to accumulate sufficient fat reserves to survive for a period of many months; in Britain, hedgehogs typically hibernate from October/November to March/April [37], although the exact timing is dependent upon a combination of both temperature and food availability [60]. Second, they need to find enough appropriate building material(s) to construct a hibernaculum that will maintain the environment within the nest at an appropriate temperature; nests are preferentially constructed from the leaves of broadleaved trees [61]. Third, the habitat must be sufficiently diverse that it offers a range of nesting locations in close proximity to one another so that an individual can relocate safely if necessary. In addition, by nesting at ground level, hedgehogs are susceptible to a range of other factors such as flooding, trampling by livestock, and disturbance by e.g., land managers, walkers and domestic dogs (*Canis familiaris*). Finally, changes in temperature patterns throughout winter may cause hedgehogs to rouse from hibernation when natural food availability is limited.

Hibernation success is, therefore, dependent on several factors, all of which may be negatively affected by agricultural intensification and/or climate change. For example: hot dry summers, soil compaction from heavy machinery and the application of pesticides and molluscicides may all reduce food availability prior to hibernation and, therefore, limit the ability of animals to acquire sufficient fat reserves to successfully complete hibernation; habitat loss and degradation may limit the number of suitable sites for hibernacula, meaning that hedgehogs may be forced to use alternative locations/habitats where preferred nesting materials are not available or where the risk of disturbance is greater; and warmer, wetter and/or more variable winters may cause animals to rouse more often and move between nests more frequently thereby depleting fat reserves and increasing susceptibility to some forms of mortality. Ultimately, such effects would be evident as: reductions in body mass before, and increased mass loss during, hibernation; an increase in the number of winter nests used and their placement in the environment; and an increase in over-winter mortality rates. These parameters would be expected to vary between areas undergoing different types of land management practice, and potentially between sexes (e.g., females may enter hibernation in poorer condition because of the energetic burden of rearing offspring, whilst males may finish hibernating earlier so that they can put on weight before the mating season).

Given the wide range of ways in which human activities could affect this phase, hibernation could represent a key critical period in the dynamics of hedgehog populations [62,63]. Despite its potential importance, little research has been conducted on the hibernation behaviour of hedgehogs in Britain in the last 40 years [37,64]. Therefore, in this study, we radio-tracked hedgehogs at one arable-dominated and one pasture-dominated site in England over the hibernation period to quantify differences in: (i) the number of winter nest sites used; (ii) patterns of habitat selection for nests; (iii) over-winter survival rates; and (iv) over-winter changes in body mass.

## 2. Materials and Methods

Data were collected from: (1) the Brackenhurst Campus (332 ha) of Nottingham Trent University, Nottinghamshire, UK (National Grid reference: SK695523); and (2) Hartpury University and College campus (339 ha), Gloucestershire, UK (National Grid reference: SO785237). Both sites were mixed commercial farms alongside a university campus, managed under the Entry level Environmental Stewardship Scheme [65]. Brackenhurst is dominated by arable fields (68.7%), with pasture fields, amenity grassland and woodland covering 24.4%, 1.9% and 2.7% of total land area, respectively. In contrast, Hartpury is dominated by pasture (34.8%) and amenity grassland (16.8%), with higher woodland (8.0%) and lower arable (30.8%) coverage than at Brackenhurst. Hedgerow length at each site is 27.1 km (Brackenhurst) and 16.9 km (Hartpury). Badgers were present at both locations: based on the numbers of setts at each site, and the frequency with which they have been photographed on motion activated trail cameras, badger density was considered comparable between the two locations. Hedgehog densities estimated in 2017 using two different methods (random encounter model based on data from trail cameras; spatial capture-recapture based on the capture history of animals along standardized transect routes) were 5.6–9.4 km^−2^ at Brackenhurst and 4.3–12.5 km^−2^ at Hartpury [66].

Fieldwork was conducted from August 2015–May 2016 and August 2016–May 2017, inclusive. Hedgehogs were captured by hand at night under licence from Natural England (ref: 20130866-0-0-0-3) using a 1-million candlepower spotlight to systematically search arable fields, pasture fields and areas of amenity grassland. Sites were surveyed at least twice per week during August and September. Once captured, animals were sexed, given a visual health check and weighed using digital scales (Salter 1035 platform scales, Salter, UK). Healthy animals weighing ≥600 g were fitted with a VHF radio transmitter (10 g: <2% of body mass; Biotrack Ltd., Wareham, UK) glued to a region of clipped dorsal spines. All animals, regardless of body mass, were marked with coloured heat shrink tubing attached to 10 dorsal spines in a unique location; tubing was attached using a portable soldering iron. The capture location was recorded with a handheld GPS unit (Garmin GPS 60, Garmin, UK). Animals were released at the point of capture, typically within 15 min.

### 2.1. Nesting Behaviour

Determining the onset of hibernation for each individual using radio-tracking is difficult. Previous authors have tended to use either a criterion based on the number of successive days a single nest was used, although these have been variable (e.g., seven days [67], one month [68]), or based upon a defined time period [64]. In this study, the latter approach was used as it was not possible to definitively identify the onset of hibernation based upon patterns of nest use alone (see Results) and because it was plausible that hibernating animals may have moved nests following e.g., disturbance by human activities.

Consequently, radio-tracking data were divided into three phases in line with the time periods defined by Yarnell et al. [64]: August–October (pre-hibernation); November–March (hibernation period); and April (post-hibernation). In the pre-hibernation phase, animals were located one night each week to record body mass and check transmitter attachment, and once per week during the day to determine the position of nests. In the hibernation phase, animals were located two-three times each week to determine the position of nests: searches were a minimum of two days apart. Radio-tracking was conducted using a Sika radio-tracking receiver and handheld, three element Yagi antenna (Biotrack).

The location of nests was recorded with a GPS unit and marked with a cane close to the nest for future identification. The position of nests was considered in the context of its specific location (e.g., in an animal burrow, hedgerow, next to or underneath a building) and the surrounding habitats (e.g., gardens, pasture, woodland). Where possible, nests were examined once they had been vacated to identify the dominant and secondary nesting materials. After examination, all nest material was left in position for future use, as hedgehogs have been found to return to nests or to occupy those of other individuals [69].

The number of nests used by each hedgehog was calculated for the time period 1 November–31 March inclusive. Where an individual had not been tracked before 1 November (n = 3) or up to 31 March (n = 3), one extra nest was added to the actual number recorded in line with the pattern of nest use observed for other animals. Differences in the number of nests used by males and females within and between the two sites were analysed using a Kruskal–Wallis test as the data were not normally distributed.

Patterns of habitat selection for winter nests were quantified by comparing the characteristics of observed (used) nest locations with those of randomly selected locations within the area available to hedgehogs. Data for each site were analysed separately. The available area was defined as the minimum convex polygon (MCP) encompassing all the diurnal and nocturnal locations from all hedgehogs radio-tracked during the study period at that site; this was used to incorporate areas outside each individual’s home range [70], and is a more objective reflection of the area used by each hedgehog population collectively than an arbitrarily predefined study area [71]. Available nest locations were randomly sampled (10 times the number of used locations) within the MCP for each study area to create an available versus used dataset. The habitat characteristics of used and available nest locations were obtained by calculating the minimum Euclidian distances to each of the seven main land cover types (amenity grassland, arable fields, buildings and associated hard-standing (hereafter ‘buildings’), hedgerows, pasture fields, roads and road verges (hereafter ‘roads’), woodland) found in both areas. All GIS analyses were carried out using ArcMap 10.3.1 software [72].

Resource Selection Functions (RSFs, [73]) based on generalised linear models for each site were used to quantify habitat selection. A logistic regression for each site was fitted, with the response variable being the used (1: GPS nesting locations) and available locations (0: random location within the MCP area defined above). Collinearity among explanatory variables was assessed using the Pearson correlation coefficient. At Brackenhurst, but not Hartpury, the minimum distances to amenity grassland and buildings were highly correlated (r = 0.7). Therefore, two different RSFs were built: Model A included amenity grassland but not buildings; Model B included buildings but not amenity grassland. Both amenity grassland and buildings were included in the Hartpury model.

Akaike’s Information Criterion (AIC) [74,75] was used for model selection. Parameter values were averaged across models within two AIC units of the best fitting model [74].

### 2.2. Patterns of Survival

Survival rates were compared between sites using Kaplan–Meier analysis [76]. Sexes and years were combined because of relatively small sample sizes (Brackenhurst n = 10; Hartpury: n = 21), and because there was no apparent difference in the number of males and females that died at each site (see Results). Because animals were captured at different times, a staggered entry [77] design was used: the first animal was captured (Day 1) on 1 August. To avoid potential biases associated with the ad hoc recovery of untagged individuals, only radio-tagged individuals were included in this analysis. Differences in survival between the two sites were quantified using a log-rank test.

### 2.3. Body Mass Changes

Differences in overwinter changes in mass were compared between sites and sexes using a series of general linear models. Mass loss was calculated using each individual’s mass at capture as close to the start and end of the hibernation period as possible; on average, animals were captured 15.5 days before 1 November and 2.6 days after 31 March. Statistical models compared differences in body mass at the start of hibernation, and mass change and percentage mass change during hibernation. All models included SITE and SEX as fixed factors and included a SEX*SITE interaction term. Linear correlation was also used to compare the number of nest sites used during hibernation with mass change over the hibernation period.

### 2.4. Data Analysis

General linear modelling and Kruskal–Wallis analyses were conducted using MINITAB version 19.1.1 and SPSS version 25, respectively. Survival analysis and RSF analyses were undertaken in R 3.3.3 [78] using lme4 and MuMIn packages [79,80]. All data were checked to ensure they conformed to the underlying assumptions of the tests used. All results are presented as mean (±SD) unless otherwise specified. As it was not possible to e.g., re-capture all tagged animals or access all nest sites, and because some animals perished during the course of the study, sample sizes vary between analyses.

## 3. Results

Forty hedgehogs were found during nocturnal surveys: 33 were fitted with radio transmitters (Table 1). Data on nesting behaviour during the hibernation period were collected from 21 hedgehogs. In total, 448 nocturnal locations, 138 nests, and 1028 diurnal locations were recorded.

### 3.1. Nesting Behaviour

The pattern of nest use was highly variable, with several animals using the same nest site for extended periods before and/or during the hibernation period (Figure 1). There was no significant difference in the number of nests used by males and females within and between the two sites (Kruskal–Wallis test: H = 0.60, DF = 3, *p* = 0.896). Combining the data, hedgehogs used a median of five nests (mean ± SD = 5.5 ± 2.3) across the 151-day hibernation period. Thirteen animals (62%) used at least one site for ≥89 days.

RSF analyses indicated that woodland, roads, pasture and, to a lesser extent, hedgerows, were consistently included in the top (ΔAIC < 2) ranked models at both sites (Figure 2; Table 2). At both sites, hedgehogs selected nest locations closer to hedgerows, in vegetation alongside roads and in woodlands, but avoided pasture fields (Table 3). Between-site differences were evident for arable fields (neither selected nor avoided at Brackenhurst; avoided at Hartpury) and both amenity grassland and buildings (both selected for at Brackenhurst in each model where these habitats were included; neither selected nor avoided at Hartpury, or not retained in top-ranked models).

At both sites, winter nests were primarily constructed from broad leaves (major component in 45% and 51% of nests, respectively: Appendix A). Major differences in the relative proportion of nests containing different materials were, however, evident. For example, litter and/or plastic waste was present in 20 nests (24%) at Hartpury, although never as the dominant material, but was never recorded at Brackenhurst.

### 3.2. Patterns of Survival

Nine animals died during the study, with no apparent sex difference in mortality risk (Brackenhurst: 1♂; Hartpury: 4♀:4♂). The overall survival rate was significantly lower at Hartpury (Log-rank test: Χ^2^_1_ = 9.46, *p* = 0.002). All deaths occurred before or after the hibernation period (Figure 3). The most common single known cause of death was predation by badgers (3 of 9 deaths; see Appendix A).

### 3.3. Body Mass Changes

Data on body mass changes across the study were available for 21 individuals. There was no significant SITE (*F*_1,17_ = 3.75, *p* = 0.069), SEX (*F*_1,17_ = 0.78, *p* = 0.389) or SITE*SEX (*F*_1,17_ = 3.75, *p* = 0.943) differences in mean body mass at the start of the hibernation period (Appendix A); collectively, hedgehogs weighed 869 ± 133 g (females: 843 ± 144 g; males: 898 ± 120 g). During hibernation, 16 individuals lost mass (Brackenhurst—5♀:3♂; Hartpury—5♀:3♂), whilst five (Brackenhurst—2♂; Hartpury—1♀:2♂) gained mass. Mass change (*F*_1,17_ = 4.65, *p* = 0.046) but not percentage mass change (*F*_1,17_ = 4.22, *p* = 0.056) differed significantly between the sexes at each site, although the latter was close to significance. At Brackenhurst, females lost 242 ± 150g on average whilst males gained a small amount of weight (4 ± 89 g; Figure 4); male and female hedgehogs at Hartpury lost 117 ± 121 g and 110 ± 141 g, respectively. These figures are equivalent to average percentage mass changes of −25%, +1%, −14% and −15%, respectively (Appendix A).

There was a negative correlation between the number of nest sites used and the loss in body mass, although this was not significant (r = −0.409, n = 21, *p* = 0.066; Figure 5). However, this was dependent on the extreme loss exhibited by a single female at Brackenhurst (432 g); excluding this female, the relationship is significant (r = −0.561, n = 20, *p* = 0.010).

## 4. Discussion

In this study, we investigated four factors associated with the winter hibernation period of hedgehogs that could potentially be affected by agricultural land-use and climate change: (i) patterns of body mass change; (ii) frequency of winter nest use; (iii) habitat selection for winter nest sites; and (iv) over-winter survival. Between the two sites studied, one dominated by arable crop production and the other by pasture and amenity grasslands, there were no apparent differences in body mass at the start of hibernation, the number of nest sites used during winter, and the selection for and avoidance of many, but not all, major habitats as nesting locations. In contrast, there were significant differences between the study sites with respect to sex-specific changes in body mass, the use of hedgerows and buildings for nesting, and patterns of survival.

### 4.1. Change in Body Mass

Estimated body mass of radio-tagged animals at the outset of the hibernation period was not significantly different between Brackenhurst and Hartpury, with animals weighing, on average 869 ± 133 g. This is likely due, in part, to the fact that we only radio-tagged individuals ≥600 g in accordance with guidance relating to the release of rehabilitated hedgehogs by the major wildlife welfare organisation in the UK [81]. This reliance on radio-tagged individuals to ensure that individuals captured before hibernation could be re-captured afterwards does, however, preclude obtaining data on animals below this threshold weight.

Acknowledging this caveat, the general pattern of mass loss observed (mean of 100–240 g within most site-sex divisions, equivalent to a mean of 14–25% of pre-hibernation mass) is within the range recorded in previous studies (Table 4). However, there was a substantial difference in sex-specific patterns of mass change at the two sites. At Hartpury, both males and females lost approximately the same amount of weight (Figure 4). Conversely, females at Brackenhurst lost markedly more weight than any other division, whereas males, on average, gained a small amount of weight. In fact, five (23.8%) animals across both sites gained weight across the hibernation period. This could indicate that individuals may have been able to access sufficient food resources during the winter period to offset the fat reserves used during hibernation, or that some animals may have already stopped hibernating and resumed typical foraging activity before they were recaptured in March/April. Although we are not able to discriminate between these possibilities, it is clear that the magnitude of these average changes are within the survivable range documented for this species.

Mass loss was also negatively correlated with the number of nests used in the winter period (Figure 5), although not significantly (*p* = 0.066). The lack of significance may, in part, be attributable to the relatively small sample size (n = 21), the highly variable changes in mass recorded, and the presence of one female that lost >400 g (40% of her body mass). Although this is among one of the largest percentage mass losses ever recorded (Table 4) and was >100 g more than any other individual in this study, this individual survived to spring. As rousing from hibernation is energetically expensive [84], hedgehogs would be expected to avoid doing so unnecessarily to avoid depleting their fat reserves. Rousing is likely to occur in response to environmental fluctuations, including both rises or falls in temperature [60], but in anthropogenic landscapes, it may also occur in response to human disturbance. To date, however, there are very few data on the extent to which disturbances affect hedgehog hibernation, either by causing them to move nests or rouse but remain in the same nest [85], and what impacts these may have on energy consumption and mortality risk.

### 4.2. Nesting Behaviour

Hedgehogs used a median of 5 (mean: 5.5) nests during the 151-day hibernation period. This is markedly higher than that observed in other studies (Table 5). Drawing direct comparisons between the number of nests used in such studies is, however, problematic because of the methodological differences used to define the onset and duration of hibernation, coupled with latitudinal differences in weather and/or temperature which extend or shorten the overall length of the hibernation period. It is worth noting, however, that the mean number of nests used by the animals in this study was more than twice that (1.74 nests per 100 days = 2.6 nests over 151 days) recorded in the most recent study of hedgehogs in England and which utilized the same dates for defining the hibernation period [64].

The increased number of nests used in our study was associated with periods during November, December and/or January where several individuals used a series of nests in quick succession (Figure 1). Although some of these periods of frequent movements between nests could be interpreted as indicating that an individual had not yet started hibernating, the patterns of nest retention exhibited throughout the study as a whole were extremely variable such that it is difficult to identify clear general trends. The possible exception to this is that the majority (62%) of animals used a single nest location for >89 days, with many of these used for the first time in November or December; this is markedly higher than the 21% of nests (n = 167) occupied for ≥3 months reported by Morris [61] in west London.

Clear patterns in nest location were evident for most, but not all, habitats. Hedgehogs consistently avoided nesting near pasture fields, whilst favouring hedgerows, woodlands and roads. In contrast, differing patterns of selection were evident for arable fields, buildings and amenity grassland. At Brackenhurst, nests were preferentially located near to amenity grassland and near buildings, although these habitats were strongly correlated with one another, whereas arable fields were neither selected nor avoided. Conversely, at Hartpury, arable fields were avoided, buildings were neither selected nor avoided and amenity grassland was not retained in the top-ranked models. These data imply that agricultural habitats were generally unsuitable for hibernation, a finding consistent with behaviour outside the hibernation period that has been attributed to a combination of reduced food availability [86] and increased risk of predation from and competition with badgers [44,45,46,47,48,49,59,87].

Hedgerows and woodland were an important habitat for nesting, a pattern that is evident in both summer and winter seasons in other studies [47,68,82,88]. Similarly, the selection for roads in this study is also most probably associated with the presence of hedgerows as borders along roads at both sites. In addition to acting as nesting sites, hedgerows are also recognised as an important refuge habitat whilst foraging where badgers are present [47,56] and for orientation through fragmented landscapes [55]. As such, the general loss and degradation of hedgerows in the UK [50,89,90] is likely to have negatively affected hedgehog populations due to impacts at multiple stages in their annual cycle, although the exact mechanisms are unknown because of the relative paucity of data on rural hedgehog populations and behaviour since the 1950s [91].

Similarly, there are few data on the importance of woodlands for hedgehogs. For example, woodlands were not identified as a factor affecting patterns of occupancy in a national survey of England and Wales [44], they were the least selected habitat in a radio-tracking study in arable landscapes [47], and no hedgehogs were detected in woodland in a pilot project on the Hartpury campus investigating the efficacy of three different methods for surveying hedgehogs [92]: all these studies were, however, conducted in the summer. The preference for woodlands as sites for hibernation observed in this study, and the reliance on broad leaves as nesting material, may suggest that hedgehogs tend to avoid woodlands during the summer months but use them as sites for hibernating during the winter months. As outlined above, one possible reason for these seasonal differences is the presence of badgers, which favour woodlands and plantations as sites for their setts [93] but undergo a period of torpor in winter [94]. Consequently, hedgehogs could be avoiding woodlands during the summer when badgers are active but using them as hibernation sites in the winter when the risk from badgers is markedly lower. As such, woodlands may represent a key resource for hedgehogs but only during one phase of their annual cycle. The impact of historical changes in the coverage of different types of woodland [95,96], their management and their interaction with an increasing badger population [57,58] on hedgehog populations are unknown but require investigation. For example, in their recent report, Mathews et al. [43] estimated that 37% of the British hedgehog population was supported by broadleaved woodland.

The affinity for amenity grassland as a foraging habitat has been well documented in Britain, most notably in the context of responses to the culling of badgers as a means for managing bovine tuberculosis in cattle [45,46,59]. During winter these areas are likely to be associated with low levels of badger activity (due to torpor) but also possibly marginally higher average temperatures than surrounding areas due to their proximity to buildings, and provision of food either accidentally (discarded refuse) or deliberately (although we were not aware of anyone deliberately feeding hedgehogs on either campus). However, amenity areas on university campuses are likely to experience high levels of pedestrian activity except in particularly poor weather and over the Christmas holiday period. The presence of buildings on these two sites also enabled hedgehogs to use some unusual nest locations, including piles of building materials and underground heating tunnels.

### 4.3. Over-Winter Survival

Survival across the study period as a whole (August–April) was significantly lower at Hartpury versus Brackenhurst. However, this was not associated with differences in mortality during the hibernation period itself, but rather mortality prior to the onset of hibernation and in the period after animals had resumed foraging in spring: in fact, none of the tagged animals in this study (n = 31) died during the hibernation period itself (Figure 3). Consequently, mortalities were not related to body mass *per se* but stochastic events such as predation by badgers and road traffic accidents (although it could be argued that animals which have not yet accumulated sufficient fat reserves and/or those that leave hibernation having lost a large amount of might be expected to take greater risks when foraging). However, it must be emphasised that these survival data are based on animals that were in good physical condition (visually health-checked and ≥600 g) prior to hibernation in accordance with welfare guidelines; this is substantially higher than the minimum threshold of 450–513 g outlined in Table 4, and which would tend to elevate survival rates.

The survival rate observed at Hartpury, when measured from August to April (approximately 65%), was lower than that recorded in Sweden (57–96%, mean = 71%) over seven years in the 1970s [62], whereas the survival rates at both sites when measured from October to April were comparable to studies from England (83%) Ireland (100%), Denmark (89–90%) and Finland (100%) conducted between 2001 and 2017 [64,67,82,97]. Overall, this body of evidence suggests that, in general terms, the survival rate of animals that have accumulated sufficient fat reserves prior to hibernation is likely to be high, but that site-specific pressures associated with movements in autumn and spring can substantially increase mortality rates [64].

## 5. Conclusions

This study has identified key similarities and differences in four key parameters associated with the winter hibernation of hedgehogs across two sites associated with different patterns of land management. Most notably, the period of hibernation itself, when hedgehogs are generally inactive within hibernacula, is not associated with high levels of mortality. Conversely, it is the periods before and after entering hibernation that pose significant risks, predominantly from stochastic factors such as badger predation and vehicle collisions. In addition, hedgehogs at both sites consistently avoided nesting in proximity to pastoral fields during winter, but favoured locations near to hedgerows, woodlands and roads. Selection for or avoidance of arable fields, buildings, and amenity grasslands varied between the two sites.

However, this study was associated with several practical limitations. Data could only be reliably collected from radio-tagged individuals and radio-tags can only be fitted to animals weighing ≥600 g for welfare reasons. Radio-tracking is also limited in the extent to which the start and end of the hibernation period (for each individual) can be identified reliably, and the ease with which data on short-term patterns of movement between nests can be collected given that animals are inactive for many successive days. Future studies, therefore, need to consider the use of other technologies, such as GPS tracking devices [98] and animal-mounted bio-loggers [99], to overcome these constraints. In particular, such studies need to focus on: (i) quantifying patterns of survival of animals weighing <600 g; (ii) identifying factors associated with nest movements and whether this affects mass change during hibernation; and (iii) the role of woodlands in the annual cycle of hedgehogs in both arable and pastoral dominated landscapes.

## Figures and Tables

**Figure 1 animals-10-01449-f001:**
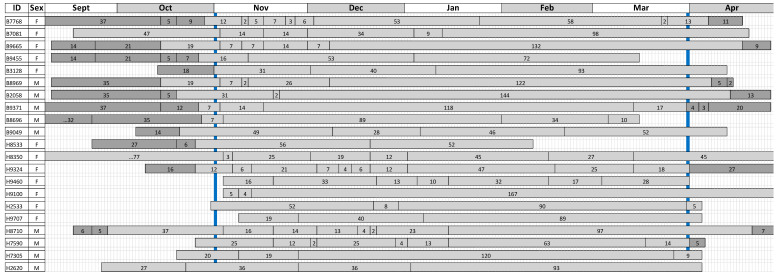
Pattern of occupation of winter nests by hedgehogs at Brackenhurst (ID numbers prefixed by “B”) and Hartpury (ID numbers prefixed by “H”). Figures in horizontal bars indicate the number of days that each nest was estimated to be occupied based upon the sampling regime (see text for details). Vertical blue columns indicate the start (1 November) and end (31 March) of the hibernation period: dark and light shaded bars indicate nests excluded from and included in the analysis of the number of nests used over the hibernation period, respectively.

**Figure 2 animals-10-01449-f002:**
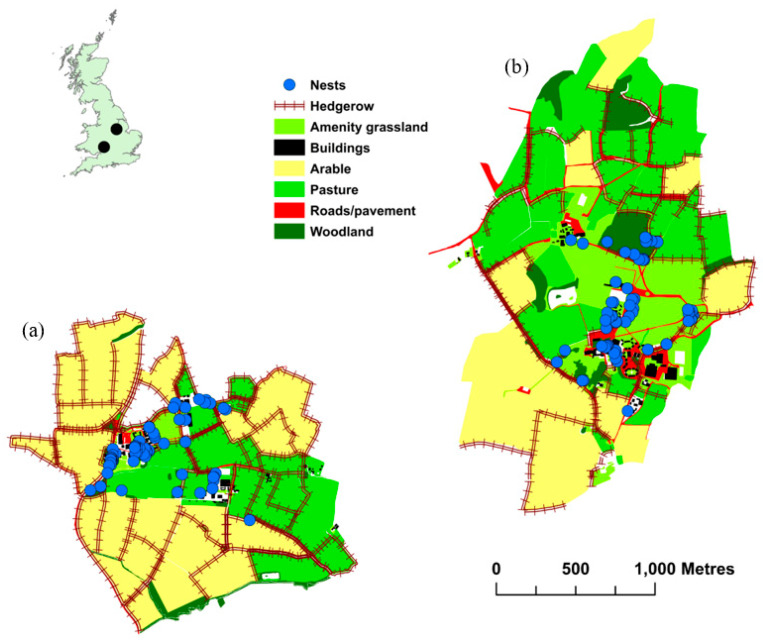
Position of hedgehog winter nest sites (blue dots) at (**a**) Brackenhurst and (**b**) Hartpury in relation to habitat composition.

**Figure 3 animals-10-01449-f003:**
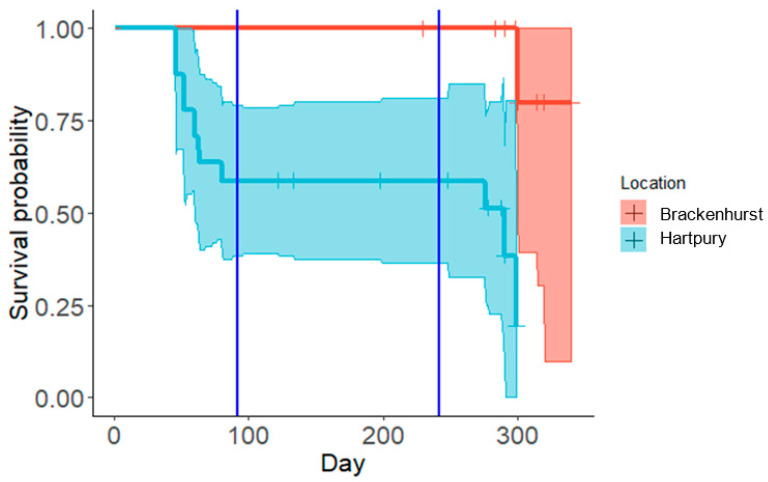
Kaplan–Meir survival functions for hedgehogs at Brackenhurst (n = 10) versus Hartpury (n = 21). Data from sexes and years (2015–2016 and 2016–2017) combined. Vertical blue lines indicate the start (1 November ) and end (31 March ) of the hibernation period.

**Figure 4 animals-10-01449-f004:**
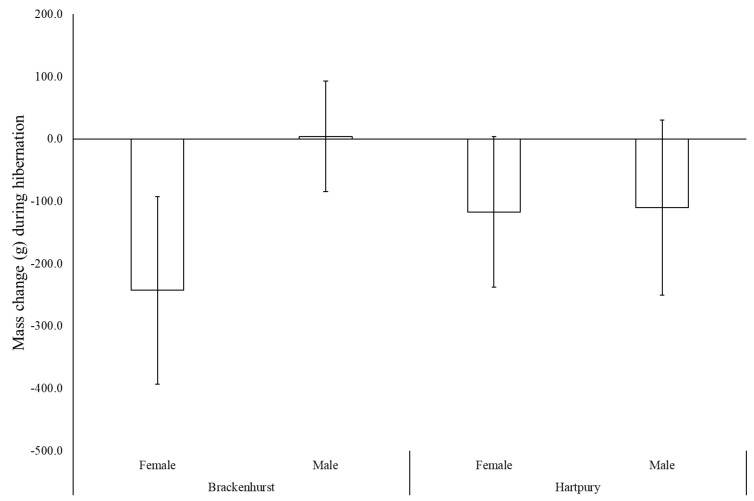
Mean (±SD) mass change during the hibernation period (1 November–31 March) in relation to site and sex (Brackenhurst: n = 5♀:5♂; Hartpury: n = 6♀:5♂).

**Figure 5 animals-10-01449-f005:**
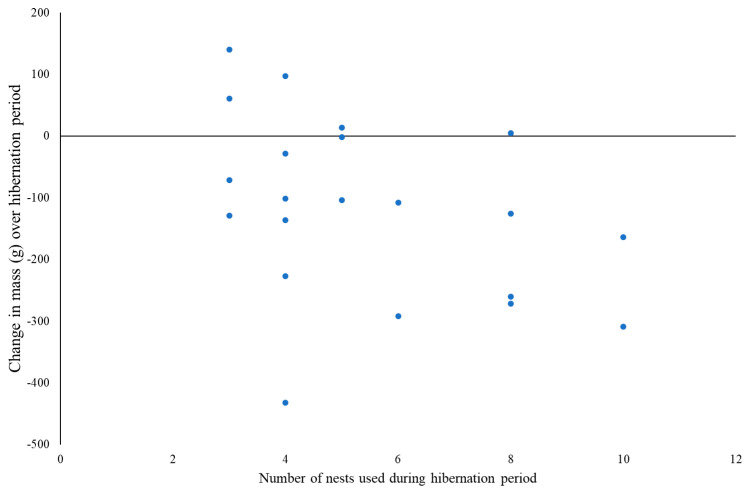
Relationship between number of nests used during the hibernation period (1 November –31 March) and the corresponding change in mass (g) over the hibernation period (n = 21).

**Table 1 animals-10-01449-t001:** Number of hedgehogs captured and radio-tagged at each site, the total number of nocturnal and diurnal locations recorded, and the number of nest sites identified.

	Brackenhurst		Hartpury		Total
	2015–2016	2016–2017	2015–2016	2016–2017	
No. captured & marked	7 (4♀:3♂)	3 (2♀:1♂)	22 (12♀:10♂)	8 (3♀: 5♂)	40 (21♀:19♂)
No. radio-tagged	7 (4♀:3♂)	3 (2♀:1♂)	18 (9♀:9♂)	5 (3♀:2♂)	33 (18♀:15♂)
No. tracked during hibernation	7 (4♀:3♂)	3 (2♀:1♂)	7 (4♀:3♂)	4 (2♀:2♂)	21 (12♀:9♂)
Total no. of nests recorded (% accessible for recording composition)	54 (59%)	12 (100%)	50 (66%)	16 (75%)	138 (65%)
No. of nocturnal locations recorded	103	74	210	61	448
No. of diurnal locations recorded	408	114	360	146	1028

**Table 2 animals-10-01449-t002:** Results of the top five *a-priori* models for predictors of habitat selection of hedgehog winter nests. Models are ranked based on their AIC values. Null model is also provided for comparison. Models indicated in bold were selected to build average models. Brackenhurst had two alternative maximal models, one including distance to amenity grassland (Brackenhurst Model A) and another including distance to buildings (Brackenhurst Model B). Habitats included in each of the top-ranking models are indicated by the “✓” symbol. Bold indicates top ranked models at each site (ΔAIC < 2).

Brackenhurst Model A									
Models (N = 64)									
Amenity grassland	Buildings	Hedgerows	Pastures	Roads	Woodland	Arable	AIC	ΔAIC	AIC_w_
✓	Not included	✓	✓	✓	✓		**357.5**	**0.00**	**0.38**
✓	Not included	✓	✓	✓	✓	✓	**358.2**	**0.75**	**0.26**
✓	Not included		✓	✓	✓	✓	**359.4**	**1.94**	**0.14**
✓	Not included		✓	✓	✓	✓	360.8	3.33	0.07
✓	Not included	✓	✓		✓	✓	362.2	4.67	0.04
NULL							491.2	134.00	<0.01
Brackenhurst Model B									
Models (N = 64)									
Amenity grassland	Buildings	Hedgerows	Pastures	Roads	Woodland	Arable	AIC	ΔAIC	AIC_w_
Not included	✓	✓	✓	✓	✓		**350.2**	**0.00**	**0.41**
Not included	✓	✓	✓	✓	✓	✓	**351.1**	**0.90**	**0.26**
Not included	✓		✓	✓	✓		**352.1**	**1.89**	**0.16**
Not included	✓		✓	✓	✓	✓	352.6	3.44	0.07
Not included	✓	✓	✓		✓	✓	354.3	4.09	0.05
NULL							491.2	141.00	<0.01
Hartpury									
Models (N = 128)									
Amenity grassland	Buildings	Hedgerows	Pastures	Roads	Woodland	Arable	AIC	ΔAIC	AIC_w_
		✓	✓	✓	✓	✓	**395.6**	**0.00**	**0.49**
	✓	✓	✓	✓	✓	✓	**397.4**	**1.80**	**0.20**
✓		✓	✓	✓	✓	✓	397.6	2.04	0.18
✓	✓	✓	✓	✓	✓	✓	399.4	3.84	0.07
		✓	✓		✓	✓	401.2	5.61	0.03
NULL							464.4	68.8	<0.01

**Table 3 animals-10-01449-t003:** Model averaged values of the best *a-priori* models (ΔAIC < 2) investigating habitat selection for winter nest sites. SE = standard error. Brackenhurst had two alternative models, one including distance to amenity grassland but excluding buildings (Brackenhurst Model A) and another including distance to buildings but excluding amenity grassland (Brackenhurst Model B). Negative values indicate a higher probability of nesting closer to that specific habitat.

Variable	Brackenhurst Model A (3 Best *a-priori* Models)				Brackenhurst Model B (3 Best *a-priori* Models)				Hartpury (2 Best *a-priori* Models)			
	Estimate	SE	z	*p*-Value	Estimate	SE	z	*p*-Value	Estimate	SE	z	*p*-Value
(Intercept)	−0.281	0.439	0.640	0.522	−0.113	0.432	0.261	0.794	−2.514	0.515	4.879	<0.001
Hedgerows	−0.013	0.006	2.000	<0.05	−0.013	0.006	2.000	<0.05	−0.008	0.003	3.204	<0.01
Pasture	0.017	0.006	2.942	<0.01	0.017	0.006	2.877	<0.01	0.010	0.003	3.748	<0.001
Roads	−0.012	0.005	2.544	<0.05	−0.010	0.004	2.443	<0.05	−0.016	0.006	2.590	<0.01
Woodland	−0.020	0.003	5.919	<0.001	−0.020	0.003	5.607	<0.001	−0.013	0.003	3.774	<0.001
Arable	0.002	0.002	1.127	0.260	0.002	0.002	1.062	0.288	0.005	0.001	3.436	<0.001
Buildings	Not included				−0.01	0.003	3.412	<0.001	0.001	0.001	0.488	0.626
Amenity grassland	−0.008	0.003	2.527	<0.05	Not included				Not included			

**Table 4 animals-10-01449-t004:** Summary of body mass changes recorded in previous studies of the West-European hedgehog over the winter hibernation period.

Country	Habitat	Years Studied	Sample Size& Composition	Mass Loss Recordedover Winter	Minimum Weightto SurviveHibernation	Reference
England	Urban parkland	1963–1968	105	25%	Recommends 450 g (550 g in more northern areas)	[63]
Denmark	Rural	2001–2002	10 (5♀:5♂); (3A:7J)	30.2 ± 7.1% (A) 22.1 ± 10.1% (J)	513 g	[82]
Ireland	Rural	2008–2009	8 (7A:1J)	301 ± 3.9g (♀) (range: 15–38%) 108 ± 2.6g (♂) (range: 3–6%)	475 g in Nov	[67]
Denmark	Suburban	2014–2015	8 (8J)	16 ± 2.9% (J)	-	[83]
England	Various	2010–2014	55 (19♀:30♂:16?); (20A:35J)	98.6 ± 35.6 g (♀) 160.8 ± 40.5 g (♂) 111.4 ± 33.0 g (A) 162.2 ± 43.3 g (J) 14.1 ± 3.1% (All animals)	Recommends >600 g for release, but one individual weighing 391 g survived release and hibernation	[64]
England	Various	2015–2017	21 (11♀:10♂)	Site 1: 240 ± 150 g (25 ± 13%) (♀) Site 1: −4 ± 89 g (1 ± 9%) (♂) Site 2: 117 ± 121 g (14 ± 16%) (♀) Site 2: 110 ± 141 g (15± 19%) (♂)	-	Present study

**Table 5 animals-10-01449-t005:** Summary of over-winter nesting behaviour in previous studies of the West-European hedgehog. Studies are listed in chronological order.

Country	Habitat	Years Studied	Sample Size& Composition ^1^	Duration of Hibernation(Days)	Number of Nests Used	Reference
England	Urban park	1963–1967	167 nests	Not recorded	Mean occupation time = 1.4 months (range 0–6 months)	[61]
Denmark	Rural	2001–2002	10 (3A:7J)	197.7 ± 2.2 (A) 178.8 ± 13.1 (J)	2.2 (range: 1–4)	[82]
Ireland	Rural	2008–2010	8 (7A:1J)	167.3 ± 10.5 (♀) 148.6 ± 10.2 (♂) 155.4 ± 9.0 (A) 157 (J)	2.0 ± 0.6 (♀) 3.2 ± 0.6 (♂) 2.4 ± 0.7 (A) 5.0 (J)	[67]
Finland	Urban	2004–2006	11 (11A) (5♀:6♂)	223 ± 2.5 (♀) 224 ± 4.8 (♂)	1.0 (♀) 1.0 (♂)	[68]
Denmark	Urban	2014–2015	8 (8J)	138.0 ± 5.6 (J)	1.8 ± 0.14 (J)	[83]
England	Various	2010–2014	55 (20A:35J); (19♀:30♂:16?)	Not recorded	2.2 ± 0.5 (♀) ^2^ 1.7 ± 0.4 (♂) ^2^ 1.8 ± 0.4 (A) ^2^ 2.6 ± 0.6 (J) ^2^	[64]
England	Arable	2015–2017	21A (12♀:9♂)	Not recorded	5.8 ± 2.6 (♀) 5.0 ± 1.9 (♂)	Present study

^1^ Data were recorded by the authors either in terms of the number of nests studied or the number of individuals studied: A = adult; J = juvenile; ? = unknown sex. ^2^ The number of nests used per 100 days.

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
