# Peer review of "Over-Winter Survival and Nest Site Selection of the West-European Hedgehog (Erinaceus europaeus) in Arable Dominated Landscapes"

_animals, 2020, doi:10.3390/ani10091449_

Round 1

Reviewer 1 Report

I really enjoyed reading this paper. I felt the manuscript was well constructed and the design and interpretation of the study was very comprehensive. It is a very valuable topic and one which I felt warranted extra research, as has been excellently done here. I look forward to reading your future work.

Hedgehogs have declined critically in the UK, with habitat loss felt to be a major cause. The ability to find habitat suitable for hibernacula is particularly important. With the rise in intensive farming and the subsequent loss of hedgerow, finding suitable sites would be expected to be more difficult. As a consequence you would expect this to be a time of significant mortality for hedgehogs. However, the current study found that contrary to what you would expect, hibernation did not prove to be a time of significant mortality and in fact no hedgehogs died at either site during this time. This was despite major individual differences in weight loss during hibernation. Habitat composition was also found to have a significant effect on the construction site of hibernacula, therefore highlighting areas of conservation concern and empathising the continued importance of hedgerows for both day nests and hibernacula.

While other studies have looked at nesting behaviour, few have looked at the survival of individuals, habitat selection, the composition of the hibernacula’s, weight loss during hibernation and its effects on post hibernation survival, duration per hibernacula or nesting behaviour in arable landscapes. There are a lot of current initiatives to encourage and improve habitats for hedgehogs so this topic is also very current and applicable, especially as the current study found that habitat composition had a significant impact on the positioning of hibernacula.

 I believe the fact that the authors have looked at the whole picture and post hibernation survival that they have gone beyond what is currently out there on nesting behaviour and winter survival in hedgehogs in the UK.

I feel that the comprehensive nature of the work makes this research convincing, the authors don’t just look at the animals during hibernation but pre and post and over a two year period. Also, the authors do not try to hide any limitations and instead highlight them with valuable advice on how things could be improved.

I feel that the authors covered all relevant aspects and in its current form I did not feel the research warranted further experiments or work.

I feel it would be interesting to look at all hedgehogs captured and not just concentrate on those that weigh 600g prior to hibernation. This would give a greater idea of vulnerable ages or pre hibernation weights and would also increase the sample size. This would not be any more work than what they have already done. If significant numbers died pre or post hibernation it would also be interesting to age them post mortem using the dentary bone. This again would highlight if particular age classes were vulnerable.

I feel the authors have discussed a broad range of previous international and national research on this topic and compared their work in an impartial and comprehensive manner. 

Author Response

REVIEWER 1

I really enjoyed reading this paper. I felt the manuscript was well constructed and the design and interpretation of the study was very comprehensive. It is a very valuable topic and one which I felt warranted extra research, as has been excellently done here. I look forward to reading your future work.

Hedgehogs have declined critically in the UK, with habitat loss felt to be a major cause. The ability to find habitat suitable for hibernacula is particularly important. With the rise in intensive farming and the subsequent loss of hedgerow, finding suitable sites would be expected to be more difficult. As a consequence, you would expect this to be a time of significant mortality for hedgehogs. However, the current study found that contrary to what you would expect, hibernation did not prove to be a time of significant mortality and in fact no hedgehogs died at either site during this time. This was despite major individual differences in weight loss during hibernation. Habitat composition was also found to have a significant effect on the construction site of hibernacula, therefore highlighting areas of conservation concern and empathising the continued importance of hedgerows for both day nests and hibernacula.

While other studies have looked at nesting behaviour, few have looked at the survival of individuals, habitat selection, the composition of the hibernacula’s, weight loss during hibernation and its effects on post hibernation survival, duration per hibernacula or nesting behaviour in arable landscapes. There are a lot of current initiatives to encourage and improve habitats for hedgehogs, so this topic is also very current and applicable, especially as the current study found that habitat composition had a significant impact on the positioning of hibernacula.

I believe the fact that the authors have looked at the whole picture and post hibernation survival that they have gone beyond what is currently out there on nesting behaviour and winter survival in hedgehogs in the UK.

I feel that the comprehensive nature of the work makes this research convincing, the authors don’t just look at the animals during hibernation but pre and post and over a two-year period. Also, the authors do not try to hide any limitations and instead highlight them with valuable advice on how things could be improved.

I feel that the comprehensive nature of the work makes this research convincing, the authors don’t just look at the animals during hibernation but pre and post and over a two-year period. Also, the authors do not try to hide any limitations and instead highlight them with valuable advice on how things could be improved.

I feel that the authors covered all relevant aspects and in its current form I did not feel the research warranted further experiments or work.

I feel it would be interesting to look at all hedgehogs captured and not just concentrate on those that weigh 600g prior to hibernation. This would give a greater idea of vulnerable ages or pre hibernation weights and would also increase the sample size. This would not be any more work than what they have already done. If significant numbers died pre or post hibernation it would also be interesting to age them post mortem using the dentary bone. This again would highlight if particular age classes were vulnerable.

Response: We agree with the point raised by the reviewer. However, as we outline in the Discussion section of the manuscript [see Lines 436-439 and 458-468], there are practical and statistical issues associated with relying on animals that have only been tagged (e.g. hedgehog spine tags) but not fitted with radio-transmitters. For example, these individuals are likely to give rise to substantial sampling biases related to (i) the likelihood that their bodies will be recovered if they died and (ii) the likelihood that they would be seen again if they survived. In the former instance, animals that died as a result of badger predation or having been run over by a vehicle are likely to be recovered, whereas those that died of starvation within a nest are not. In the case of the latter, animals that survived but moved off of the study site would be very unlikely to be recaptured. As a consequence of these limitations, we were reliant on animals fitted with radio-transmitters such that we could identify the definitive fate of each animal. But, as we highlight, this does impose significant limitations: most notably, for welfare reasons, we could only fit transmitters to animals weighing 600g or more; and we acknowledge that this is likely to be associated with elevated survival rates [see Lines 436-439] and that this is something that future studies need to address [see Lines 465-466] so that we can get a more complete picture of over-winter mortality in this species. This will require the use of novel field methods so that unbiased data can be collected from animals weighing <600g.

Given the fact that any data from animals weighing <600g would be associated with marked biases, and that actually little such data were collected at each of the two study sites (i.e. very few animals weighing less were captured and tagged in the first place; and that many of these were subsequently re-captured) we do not think that adding this information into the manuscript would reflect sound practice.

I feel the authors have discussed a broad range of previous international and national research on this

Reviewer 2 Report

The authors have offered a significant addition to the scarce literature on hedgehog winter survival. Overall, the manuscript is very well written and the data very well analysed, presented and discussed. Therefore, I recommend its publication in Sustainability after a couple of corrections.

Comments

Lines 114-115 – Correct either percentages or characterization for woodland and arable coverage. 8.0% is not higher than 30.8%.

Lines 285-288, 337, Figures 4 and 5. While the authors might mean that the change in mass was either positive or negative, their use of “absolute mass loss” or absolute mass change” is not correct mathematically and is also confusing. Although I finally got it, I was confused at first. If authors had used absolute change, they would have provided positive only numbers. In doing so, the change in mass (which could have been positive or negative), in absolute number terms, would be positively correlated with the number of nests used. Just remove “absolute” to read “mass loss” or “mass change”.

Author Response

REVIEWER 2

The authors have offered a significant addition to the scarce literature on hedgehog winter survival. Overall, the manuscript is very well written and the data very well analysed, presented and discussed. Therefore, I recommend its publication in Sustainability after a couple of corrections.

Comments

Lines 114-115 – Correct either percentages or characterization for woodland and arable coverage. 8.0% is not higher than 30.8%

Our reply: This appears to be a misunderstanding on the point of the reviewer. In the manuscript, [see Lines 112-115] we state the following

“Brackenhurst is dominated by arable fields (68.7%), with pasture fields, amenity grassland and woodland covering 24.4%, 1.9% and 2.7% of total land area, respectively. In contrast, Hartpury is dominated by pasture (34.8%) and amenity grassland (16.8%) with higher woodland (8.0%) and lower arable (30.8%) coverage”

The figures quoted for woodland and arable highlighted by the reviewer relate to comparisons between the Hartpury site relative to the Brackenhurst site, rather than within the Hartpury site. For example, the 8.0% woodland coverage at Hartpury is higher than the 2.7% at Brackenhurst; and the 30.8% coverage at Hartpury is lower than the 68.7% at Brackenhurst. However, given that this was a point of confusion, we have amended the text slightly to make this point clearer [see Lines 113-115]

Lines 285-288, 337, Figures 4 and 5. While the authors might mean that the change in mass was either positive or negative, their use of “absolute mass loss” or absolute mass change” is not correct mathematically and is also confusing. Although I finally got it, I was confused at first. If authors had used absolute change, they would have provided positive only numbers. In doing so, the change in mass (which could have been positive or negative), in absolute number terms, would be positively correlated with the number of nests used. Just remove “absolute” to read “mass loss” or “mass change”

Response: We have amended the text and the Figures in line with the point raised by the reviewer. In addition, to the locations in the text outlined by the reviewer, we have also amended the text at several other places so that the term “absolute mass change” does not appear at all. These amendments, which include deletions as well as changes in the wording, were made on Lines 199, 200, 282, 289, 297, 303, 341 and 478